# *Pseudomonas aeruginosa* Bloodstream Infections Presenting with Septic Shock in Neutropenic Cancer Patients: Impact of Empirical Antibiotic Therapy

**DOI:** 10.3390/microorganisms12040705

**Published:** 2024-03-30

**Authors:** Cristina Royo-Cebrecos, Júlia Laporte-Amargós, Marta Peña, Isabel Ruiz-Camps, Carolina Garcia-Vidal, Edson Abdala, Chiara Oltolini, Murat Akova, Miguel Montejo, Malgorzata Mikulska, Pilar Martín-Dávila, Fabián Herrera, Oriol Gasch, Lubos Drgona, Hugo Manuel Paz Morales, Anne-Sophie Brunel, Estefanía García, Burcu Isler, Winfried V. Kern, Zaira R. Palacios-Baena, Guillermo Maestr de la Calle, Maria Milagro Montero, Souha S. Kanj, Oguz R. Sipahi, Sebnem Calik, Ignacio Márquez-Gómez, Jorge I. Marin, Marisa Z. R. Gomes, Philipp Hemmatii, Rafael Araos, Maddalena Peghin, Jose L. Del Pozo, Lucrecia Yáñez, Robert Tilley, Adriana Manzur, Andrés Novo, Jordi Carratalà, Carlota Gudiol

**Affiliations:** 1Internal Medicine Department, Hospital Nostra Senyora de Meritxell, SAAS, AD700 Escaldes-Engordany, Andorra; cristina.royoceb@gmail.com; 2Infectious Diseases Department, Bellvitge University Hospital, IDIBELL, 08907 Barcelona, Spain; jcarratala@bellvitgehospital.cat; 3Haematology Department, Institute Català d’Oncologia (ICO)–Hospital Duran i Reynals, IDIBELL, 08908 Barcelona, Spain; mpena@iconcologia.net; 4Infectious Diseases Department, Vall d’Hebron University Hospital, 08035 Barcelona, Spain; isabelruizcamps@gmail.com; 5Infectious Diseases Department, Hospital Clínic i Provincial, 08036 Barcelona, Spain; carolgv75@hotmail.com; 6Instituto do Cancer do Estado de São Paulo, Faculty of Medicine, Univesity of São Paulo, Sao Paulo 01246, Brazil; eabdala@uol.com.br; 7Unit of Infectious and Tropical Diseases, IRCCS San Raffaele Scientific Institute, 20132 Milan, Italy; chiara.oltolini@ospedaleniguarda.it; 8Department of Infectious Diseases, Hacettepe University School of Medicine, 06100 Ankara, Turkey; akova.murat@gmail.com; 9Infectious Diseases Unit, Cruces University Hospital, 48903 Bilbao, Spain; josemiguelmontejo@hotmail.com; 10Division of Infectious Diseases, Ospedale Policlinico San Martino, University of Genoa (DISSAL), 16132 Genoa, Italy; m_mikulska@yahoo.com; 11Infectious Diseases Department, Ramon y Cajal Hospital, 28034 Madrid, Spain; pmartindav@gmail.com; 12Infectious Diseases Section, Department of Medicine, Centro de Educación Médica e Investigaciones Clínicas (CEMIC), Buenos Aires C1430EFA, Argentina; fabian1961@gmail.com; 13Infectious Diseases Department, Hospital Universitari Parc Taulí, Institut d’Investigació i Innovació Parc Taulí (I3PT-CERCA), Universitat Autònoma de Barcelona, 08208 Sabadell, Spain; ogasch@tauli.cat; 14Oncohematology Department, National Cancer Institute, Comenius University, 81499 Bratislava, Slovakia; lubos.drgona@gmail.com; 15Infectious Diseases Department, Hospital Erasto Gaertner, Curitiba 81520-060, Brazil; moraleshmp@gmail.com; 16Infectious Diseases and Medicine Department, Lausanne University Hospital, CHUV, 1011 Lausanne, Switzerland; anne-sophie.brunel@chuv.ch; 17Haematology Department, Reina Sofía University Hospital-IMIBIC-UCO, 14004 Córdoba, Spain; egarcia@gmail.com; 18Department of Infectious Diseases and Clinical Microbiology, Istanbul Education and Research Hospital, 34668 Istanbul, Turkey; burcubayrak85@gmail.com; 19Division of Infectious Diseases, Department of Medicine II, Faculty of Medicine, University of Freiburg Medical Center, 79110 Freiburg, Germany; winfried.kern@uniklinik-freiburg.de; 20Unit of Infectious Diseases and Clinical Microbiology, Institute of Biomedicine of Seville (IBIS), Virgen Macarena University Hospital, 41013 Seville, Spain; zaira.palacios.baena@hotmail.com; 21Infectious Diseases Unit, Instituto de Investigación Hospital “12 de Octubre” (i + 12), School of Medicine, “12 de Octubre” University Hospital, Universidad Complutense, 28041 Madrid, Spain; gmaestro@gmail.com; 22Infectious Pathology and Antimicrobials Research Group (IPAR), Infectious Diseases Service, Hospital del Mar, Institut Hospital del Mar d’Investigations Mèdiques (IMIM), Universitat Autònoma de Barcelona (UAB), CEXS-Universitat Pompeu Fabra, 08003 Barcelona, Spain; mmontero@psmar.cat; 23Infectious Diseases Division, American University of Beirut Medical Center, Beirut 110236, Lebanon; sk11@aub.edu.lb; 24Faculty of Medicine, Ege University, 35040 Izmir, Turkey; oguz.resat.sipahi@gmail.com; 25Department of Infectious Diseases and Clinical Microbiology, University of Health Science Izmir Bozyaka Training and Research Hospital, 35170 Izmir, Turkey; sebnemozkoren@yahoo.com; 26Infectious Diseases Department, Hospital Regional de Málaga, 29010 Málaga, Spain; marquezgomez@gmail.com; 27Infectious Diseases and Clinical Microbiology Department, Clínica Maraya, Manizales 170001-17, Colombia; jimarin.uribe@gmail.com; 28Hospital Federal dos Servidores do Estado, Instituto Oswaldo Cruz, Fundação Oswaldo Cruz, Ministério da Saúde, Rio de Janeiro 20221-161, Brazil; mzrgomes@gmail.com; 29Department of Hematology, Oncology and Palliative Care, Klinikum Ernst von Bergmann, Academic Teaching Hospital of Charité University Medical School, 10117 Berlin, Germany; philipp@hemmati.de; 30Instituto de Ciencias e Innovación en Medicina, Facultad de Medicina Clínica Alemana Universidad del Desarrollo, Santiago de Chile 12461, Chile; rafaaraos@gmail.com; 31Infectious and Tropical Diseases Unit, Department of Medicine and Surgery, University of Insubria-ASST-Sette Laghi, 21100 Varese, Italy; maddalena.peghin@gmail.com; 32Infectious Diseases and Microbiology Unit, Navarra University Clinic, 31008 Pamplona, Spain; jdelpozo@unav.es; 33Haematology Department, Marqués de Valdecilla University Hospital, 39008 Santander, Spain; lucrecia22176@icloud.com; 34Microbiology Department, University Hospitals Plymouth NHS Trust, Plymouth PL6 8DH, UK; robert.tilley@nhs.net; 35Infectious Diseases, Hospital Rawson, San Juan J5400, Argentina; manzuradriana@gmail.com; 36Haematology Department, Son Espases University Hospital, 07120 Palma de Mallorca, Spain; andres.novo61@gmail.com; 37Faculty of Medicine, Bellvitge Campus, University of Barcelona, carrer de la Feixa Llarga, s/n, 08907 Barcelona, Spain; 38Centro de Investigación Biomédica en Red de Enfermedades Infecciosas (CIBERINFEC), Instituto de Salud Carlos III, 28029 Madrid, Spain; 39Infectious Diseases Unit, Catalan Institute of Oncology (ICO), Duran i Reynals Hospital, IDIBELL, 08908 Barcelona, Spain

**Keywords:** *Pseudomonas aeruginosa*, bacteremia, septic shock, bloodstream infection, neutropenia, cancer

## Abstract

This large, multicenter, retrospective cohort study including onco-hematological neutropenic patients with *Pseudomonas aeruginosa* bloodstream infection (PABSI) found that among 1213 episodes, 411 (33%) presented with septic shock. The presence of solid tumors (33.3% vs. 20.2%, *p* < 0.001), a high-risk Multinational Association for Supportive Care in Cancer (MASCC) index score (92.6% vs. 57.4%; *p* < 0.001), pneumonia (38% vs. 19.2% *p* < 0.001), and infection due to multidrug-resistant *P. aeruginosa* (MDRPA) (33.8% vs. 21.1%, *p* < 0.001) were statistically significantly higher in patients with septic shock compared to those without. Patients with septic shock were more likely to receive inadequate empirical antibiotic therapy (IEAT) (21.7% vs. 16.2%, *p* = 0.020) and to present poorer outcomes, including a need for ICU admission (74% vs. 10.5%; *p* < 0.001), mechanical ventilation (49.1% vs. 5.6%; *p* < 0.001), and higher 7-day and 30-day case fatality rates (58.2% vs. 12%, *p* < 0.001, and 74% vs. 23.1%, *p* < 0.001, respectively). Risk factors for 30-day case fatality rate in patients with septic shock were orotracheal intubation, IEAT, infection due to MDRPA, and persistent PABSI. Therapy with granulocyte colony-stimulating factor and BSI from the urinary tract were associated with improved survival. Carbapenems were the most frequent IEAT in patients with septic shock, and the use of empirical combination therapy showed a tendency towards improved survival. Our findings emphasize the need for tailored management strategies in this high-risk population.

## 1. Introduction

Bloodstream infection (BSI) is a major cause of morbidity and mortality in neutropenic cancer patients. A shift in the etiology of BSI to Gram-negative bacilli (GNB) as well as an increase in antibiotic resistance has been reported in neutropenic cancer patients in recent decades [1,2,3,4]. Among these pathogens, *Pseudomonas aeruginosa* has historically been highlighted as a major contributor to severe sepsis and increased mortality in neutropenic cancer patients [5,6,7,8]. Thus, the emergence of antimicrobial resistance in *P. aeruginosa* is of special concern since inadequate empirical antibiotic therapy (IEAT) is associated with increased mortality in this setting [9,10,11,12]. *P. aeruginosa* is also well known for its ability to form biofilms, which significantly contribute to chronic infections and septic episodes [13,14]. These biofilms serve as protective environments, enhancing bacterial survival and resistance to host immune defenses as well as antimicrobial treatments [15,16].

Despite the above-mentioned epidemiological changes, the most up-to-date antibiotic guidelines for the treatment of febrile neutropenia in cancer patients continue to recommend an empirical β-lactam treatment that includes cefepime, piperacillin–tazobactam, and meropenem [17,18,19]. Among patients administered this treatment, some authors have reported high rates of IEAT in neutropenic cancer patients with BSI. Chumbita et al. recently published the results of a multicenter study involving 700 episodes of BSI in neutropenic hematological patients, in which they found that 34.7% of the isolated GNB were resistant to at least one of the three β-lactams recommended in febrile neutropenia guidelines [20]. Among episodes caused by GNB, 16.6% received IEAT, and this rate increased to 46.3% when the infection was caused by an MDRGNB. The same authors published a retrospective study involving 280 episodes of PABSI in hematologic neutropenic patients, in which 36.1% of the isolates were also resistant to at least one of the β-lactam antibiotics recommended in these international guidelines [21]. In addition, even when international guidelines were widely followed, 16.8% of patients received IEAT and 23.6% received inadequate β-lactam empirical antibiotic treatment.

Septic shock is an uncontrolled systemic response to infection that can quickly lead to multiple organ dysfunction and, ultimately, death if not promptly and effectively managed [22]. The need for immediate and targeted treatment lies in the patient’s survival and in mitigating long-term sequelae [23]. Furthermore, the proper treatment of infections and septic shock in neutropenic patients contributes to preserving the integrity of the weakened immune system, which is essential for ongoing cancer management and the patient’s successful recovery. Therefore, early identification, prompt adequate empirical antimicrobial therapy, and efficient hemodynamic support are fundamental pillars in caring for these vulnerable patients, aiming to improve their survival rates and long-term quality of life [24]. 

There is scarce information regarding the clinical features and outcomes of neutropenic patients with BSI presenting with septic shock. In a large study involving 1563 neutropenic patients with BSI, 257 (16%) presented with septic shock, and the great majority of the infections were caused by GNB (81%) [25]. IEAT (17.5%) was identified as an independent risk factor for mortality, whereas the empirical combination of β-lactam and amikacin was found to be protective.

The current study aimed to assess the clinical features and outcomes of *P. aeruginosa* bloodstream infection (PABSI) in neutropenic patients presenting with septic shock, identify the risk factors for 30-day case fatality rate, and evaluate the impact of IEAT.

## 2. Material and Methods

### 2.1. Study Design and Setting

The current study is part of the IRONIC project: a large, retrospective, multicenter, international cohort study conducted at 34 centers in 12 countries from 1 January 2006 to 31 May 2018. The number of participating centers and the number of patients recruited at each one have been published previously [26].

### 2.2. Participants

The study included adult patients (≥18 years) with solid tumors and hematological diseases, including hematopoietic stem cell transplant recipients, who suffered from at least one episode of PABSI during the study period. Each episode of PABSI identified in the same patient was considered and documented as a distinct episode if it occurred after an interval exceeding four weeks from the previous episode. Patients were followed for the following 30 days from BSI onset.

### 2.3. Variables

Data regarding baseline and epidemiological characteristics, clinical features, microbiological findings, and outcomes were recorded. Empirical antibiotic therapy was defined as the antibiotic administered before the definitive susceptibility results were received. Adequate empirical antibiotic therapy was considered when patients received at least one in vitro active antibiotic against the *P. aeruginosa* isolate. IEAT was considered when the patient did not receive any empirical antibiotic with in vitro activity or an empirical antibiotic therapy was lacking. In addition, in patients with pneumonia, empirical monotherapy with an aminoglycoside was considered inadequate. The antibiotics were uniformly administered at the current standard doses for the treatment of febrile neutropenia [17,19]. In case of renal impairment, the doses were adjusted accordingly.

### 2.4. Outcomes

Episodes of PABSI occurring in patients with septic shock were compared to those developing in patients without septic shock. Risk factors associated with overall 30-day case fatality rate were investigated in patients with septic shock.

### 2.5. Microbiological Studies

Clinical samples were processed at the microbiology laboratories of each participating center by standard operating procedures. *P. aeruginosa* was identified using standard microbiological techniques at each center. In vitro susceptibility was determined according to the EUCAST recommendations [27], except one center from Argentina, the Lebanese center, where the CLSI breakpoints were used, and the center in the UK where the BSAC recommendations were used before 2016. *P. aeruginosa* isolate phenotypes were classified as multidrug-resistant (MDR) and extensively resistant (XDR) following the definitions by Magiorakos et al. [28].

### 2.6. Definitions

Neutropenia and severe neutropenia were defined as an absolute neutrophil count below 0.5 × 10^9^ cells/mm and 0.1 × 10^9^ cells/mm, respectively. Septic shock was considered as a clinically defined subset of sepsis cases, wherein, despite adequate fluid resuscitation, patients had hypotension requiring vasopressors to maintain a mean arterial blood pressure above 65 mm Hg and had an elevated serum lactate concentration of more than 2 mmol/L [22,29]. The Multinational Association for Supportive Care in Cancer (MASCC) score was calculated as described previously [30]. Prior corticosteroid treatment was defined as the administration of ≥20 mg of prednisone, or equivalent dosing, for at least 4 weeks within 30 days of BSI onset.

Comorbidities were defined as the presence of one or more of the following diseases: diabetes mellitus, chronic heart disease, chronic obstructive pulmonary disease, chronic renal disease, chronic hepatic disease, and cerebrovascular disease.

BSI sources were established using standard US Centers for Disease Control and Prevention criteria [31]. Additionally, an endogenous source of BSI was considered in neutropenic patients with mild or absent gastrointestinal symptoms, in whom gut translocation was suspected. Patients with severe (grade III–IV) extensive mucositis involving the upper and lower gastrointestinal tract were considered to present neutropenic enterocolitis. Mucositis was diagnosed in patients with ulcerative lesions involving only the oral cavity. Patients with no clear source of BSI due to mixed clinical presentations and/or lack of adequate complementary diagnostic procedures were considered to have an unknown source of BSI.

BSI was considered to be persistent if blood cultures remained positive after 48 h of adequate antibiotic therapy. The 7-day and 30-day case fatality rates were defined as death from any cause within 7 days and 30 days of BSI onset, respectively.

### 2.7. Statistical Analysis

To characterize the cohort, categorical variables were represented as case counts and percentages, while continuous variables were summarized as either mean and standard deviation (SD) or median and interquartile range (IQR). We employed Student’s t-test or the Mann–Whitney U test, as appropriate, for comparing continuous variables. For assessing the relationship between categorical variables, Fisher’s exact test or Pearson’s χ^2^ test was utilized. We calculated odds ratios (ORs) along with their corresponding 95% confidence intervals (CIs). Statistical significance was determined at a threshold of *p* < 0.05. The statistical analysis was carried out using the stepwise logistic regression model within the SPSS software package, version 27.0 (SPSS Inc., Chicago, IL, USA).

### 2.8. Ethics

The study was approved by the Institutional Review Board at Bellvitge University Hospital [local reference number PR408/17] and by the local research ethics committees at the participating centers. It was conducted in accordance with the Declaration of Helsinki guidelines. The need for informed consent was waived by the Clinical Research Ethics Committee due to the study’s retrospective design.

## 3. Results

### 3.1. Clinical Characteristics

Of 1217 episodes of PABSI, 1213 were eligible for analysis, of which 411 (33.8%) presented with septic shock. The epidemiological and clinical characteristics of patients included in the study are detailed in Table 1. Septic shock was significantly more common in males than in females (66.7% vs. 59.2%; *p* = 0.012), in patients with solid tumors compared with hematologic malignancies (33.2% vs. 20.2%; *p* < 0.001), in patients with a high-risk MASCC index score (92.6% vs. 57.4%, *p* < 0.001), in patients with a urinary catheter in place (26% vs. 12.8%; *p* < 0.001), and in patients with BSI acquired outside the hospital (59.2% vs. 52.6%, *p* = 0.006). Pneumonia as the source of BSI was significantly associated with septic shock (38% vs. 19.2% *p* < 0.001), whereas endogenous and perineal BSIs were less likely to present with septic shock (41% vs. 29.9%; *p* < 0.001, and 3.6% vs. 1.2% *p* < 0.001, respectively). Infections due to MDRPA and XDRPA were more frequently associated with septic shock (33.8% vs. 21.1.1%, *p* < 0.001, and 27.5% vs. 15.3%, *p* < 0.001, respectively).

### 3.2. Initial Empirical Antibiotic Therapy and Clinical Outcomes

Table 2 displays the initial empirical antibiotic therapy and clinical outcomes of patients compared by groups. Patients with septic shock were more likely to receive empirical combination therapy (45.3% vs. 34.9%; *p* < 0.001), mainly β-lactam plus aminoglycoside, whereas patients without septic shock received monotherapy more frequently (65.1 vs. 54.7%, *p* < 0.001). The rate of IEAT was significantly higher in patients with septic shock (21.7% vs. 16.2%; *p* = 0.020) and in those with infection due to MDRPA and XDRPA strains (68% vs. 16%; *p* < 0.001, and 54.8% vs. 11.6%; *p* < 0.001, respectively). The antibiotics most frequently deemed inadequate were carbapenems and piperacillin–tazobactam in both groups. Patients with septic shock presented poor outcomes, with a higher need for intensive care unit (ICU) admission and mechanical ventilation (74% vs. 10.5%, *p* < 0.001, and 49.1% vs. 5.6%, *p* < 0.001, respectively) and higher rates of persistent BSI (18.3% vs. 7.7%; *p* < 0.001). Overall, the 7-day and 30-day case fatality rates were 27.7% and 40.3%, which were significantly higher in patients with septic shock than in those without (58.2% vs. 12.1%, *p* < 0.001, and 74% vs. 23.1%, *p* < 0.001, respectively).

### 3.3. Risk Factors Associated with Overall Case Fatality Rate

The risk factors associated with 30-day mortality are described in Table 3. Mechanical ventilation (OR 9; 95% CI 4.72–17.18; *p* < 0.001), infection due to MDRPA (OR 1.98; 95% CI 1.00–3.90; *p* = 0.048), IEAT (OR 2.47; 95% CI 1.01–5.88; *p* = 0.047), and persistent BSI (OR 2.49; 95% CI 1.12–5.56; *p* = 0.026) were independently associated with increased mortality. Conversely, BSI from the urinary tract (OR 0.27; 95% CI 0.07–0.99; *p* = 0.047) and the administration of granulocyte colony-stimulating factor (OR 0.33; 95% CI 0.19–0.58; *p* < 0.001) were independently associated with improved survival. Empirical combination therapy showed a tendency toward lower mortality.

## 4. Discussion

This extensive, international, multicenter cohort of neutropenic cancer patients with PABSI showed that septic shock was frequent in this setting and that it was more common in patients with solid tumors and in those with pneumonia. The rates of multidrug resistance and the administration of IEAT were higher in patients with septic shock and were associated with poor outcomes, including mechanical ventilation and early 7-day and overall 30-day case fatality rates.

Septic shock was more frequent in males, in patients with solid tumors, and in those with pneumonia [32]. These findings are in line with previous research that suggests a gender-based discrepancy in infection-related outcomes and tumor-specific immune response profiles [33,34,35]. The most frequent underlying solid tumor in our cohort was lung cancer, which is more prevalent in males and is a well-known risk factor for *P. aeruginosa* pneumonia [36,37,38]. Kang et al. established a notable correlation between pneumonia and severe sepsis or septic shock, finding a 2.7-fold increase in pneumonia in the severe-sepsis group, particularly when GNB were involved [38]. More recently, our group identified an increased risk of septic shock and death in patients with bacteremic *P. aeruginosa* pneumonia compared to other sources of PABSI [36].

Of note, infection due to MDRPA and XDRPA was significantly higher in patients with septic shock. Furthermore, multidrug resistance was independently associated with increased overall 30-day case fatality rate in this group of patients. This is of special concern because antibiotic resistance has remarkably increased in recent decades among isolates of *P. aeruginosa* causing severe infections in neutropenic cancer patients, and the administration of adequate empirical antibiotic therapy in this setting may be compromised [8,11]. Future research could explore the interplay between antimicrobial resistance at the single-cell level and within biofilms not addressed in our study, building upon previous studies to elucidate comprehensive mechanisms of resistance [39,40,41].

Importantly, we found that patients with septic shock were more likely to receive IEAT and, additionally, IEAT was identified as an independent risk factor for 30-day mortality. It is important to highlight that the most frequent inadequate antibiotics prescribed in patients with septic shock were carbapenems (40.4%). This finding underscores the rationale for using other non-carbapenem β-lactams empirically, including the more recently commercially available ceftolozane–tazobactam and cefiderocol in selected patients [42,43,44].

In the current era of widespread antimicrobial resistance, the potential benefit of using empirical combination therapy gains importance. In this regard, a recent study involving neutropenic hematological patients with Gram-negative BSI found that empirical therapy with a broad-spectrum β-lactam plus a short-course aminoglycoside regimen significantly improved the 7-day case fatality rate compared to β-lactam monotherapy, with no significant renal function impairment [45]. In this study, the overall rate of septic shock was 19.3% and *P. aeruginosa* was the second etiological agent of BSI. Chumbita et al. reported a high mortality rate (54.9%) in neutropenic patients with BSI who received IEAT, whilst reporting that the empirical use of a β-lactam plus amikacin was protective against a fatal outcome [25]. Additionally, in research included in the IRONIC project that analyzed the episodes of bacteremic pneumonia due to *P. aeruginosa*, we found that IEAT was an independent risk factor for increased 30-day case fatality, whereas the use of an appropriate empirical combination treatment was associated with improved survival [36]. In the current study, the empirical combination of a β-lactam plus an aminoglycoside showed a tendency towards a lower 30-day case fatality rate (*p* = 0.072).

In addition to infection due to MDRPA and IEAT, mechanical ventilation and persistent BSI were also risk factors for mortality [46]. Patients requiring ICU admission and mechanical ventilation are critically ill patients who may develop life-threatening nosocomial complications. Interestingly, prior studies have reported lower ICU admission rates and less need for mechanical ventilation in patients with cancer [37,47,48]. These data raise a pertinent question regarding whether a more proactive and assertive approach to medical management of cancer patients with infectious complications, and particularly in cases of pneumonia, could have potentially resulted in better outcomes for these individuals. The high mortality rates observed in patients admitted to the ICU and the generally unfavorable long-term prognosis associated with malignancy have led to the perception that cancer patients may not be suitable candidates for ICU admission. Intriguingly, as many as 50% of cancer patients referred for ICU admission are finally not admitted, with clinicians citing them as either too healthy or too critically ill to derive significant benefits [49]. Nevertheless, there is growing evidence that survival rates have notably improved in cancer patients who have experienced septic shock and pneumonia, suggesting that the presence of an underlying malignancy may no longer be the sole determinant of their prognosis [46,50,51,52]. In addition, another study revealed that in-hospital and ICU mortality rates, as well as the length of stay, did not significantly differ between septic shock patients with and without cancer [32]. These findings suggest that malignancies should no longer be seen as a barrier to ICU admission for this patient population [32,52]. Persistent BSI involves the inability to eradicate bacteria from the bloodstream, primarily due to intravascular or high-inoculum infections, or due to immune system defects, such as neutropenia.

Conversely, BSI from the urinary tract and the use of granulocyte colony-stimulating factor (G-CSF) emerged as protective factors in patients with septic shock. This underscores the critical role of neutrophil recovery in improving survival rates. The use of G-CSF in neutropenic hematological patients remains controversial due to the fear of triggering an uncontrolled blastic response [19]. However, in patients with life-threatening complications, such as BSI with septic shock, it seems reasonable to use G-CSF as adjuvant therapy along with prompt antibiotic treatment and adequate hemodynamic support [53,54,55].

The main strength of this study is that it is based on one of the largest cohorts of neutropenic cancer patients with PABSI with a multicenter international design that allows the generalization of the results. Nevertheless, this study also has some limitations that should be acknowledged. The retrospective design and the inherent heterogeneity of the patient population may confound the associations observed. Secondly, the absence of randomization in this clinical study means that therapeutic selections may have been influenced by patient-related variables and by the clinical presentation. Ultimately, given the extended duration of the study, we cannot discount the potential influence of time-related factors on certain variables, such as mortality rates.

In conclusion, septic shock in neutropenic patients with PABSI was frequent, and it was more common in patients with solid tumors and in those with pneumonia. Multidrug resistance and IEAT were more frequent in these patients and were associated with poor outcomes. Carbapenems were the most frequent IEAT in patients with septic shock, and the use of empirical combination therapy showed a tendency towards improved survival. Our findings emphasize the need for tailored management strategies in this high-risk population.

## Figures and Tables

**Table 1 microorganisms-12-00705-t001:** Epidemiological and clinical characteristics of episodes of *P. aeruginosa* bloodstream infection in patients with and without septic shock.

Characteristics	Overall *n* = 1213	No Septic Shock *n* = 802 (%)	Septic Shock *n* = 411 (%)	*p*-Value
Age (years, median, range)	60 (IQR 20)	60 (IQR 21)	60 (IQR 19)	0.423
Male sex	749	475 (59.2)	274 (66.7)	0.012
Co-morbidities				
Chronic heart disease	150	94 (11.7)	56 (13.6)	0.340
Chronic obstructive pulmonary disease	118	71 (8.9)	47 (11.4)	0.151
Diabetes mellitus	137	99 (12.3)	38 (9.2)	0.107
Chronic liver disease	85	54 (6.7)	31 (7.5)	0.601
Other comorbidities ^a^	146	102 (13.3)	44 (11.4)	0.372
Solid tumors	299	162(20.2)	137 (33.3)	<0.001
Lung cancer	88	50 (30.9)	38 (27.7)	
Gastrointestinal cancer	52	25(15.4)	27 (19.7)	
Breast cancer	28	18 (11.1)	10 (7.3)	
Genitourinary cancer	29	11 (6.8)	18 (13.1)	
Other solid tumors	102	58 (35.8)	44 (32.1)	
Hematologic malignancies	914	640 (79.8)	274 (66.7)	<0.001
Acute myeloid leukemia	310	233 (36.4)	77 (28.1)	
Non-Hodgkin lymphoma	271	179 (28)	92 (33.6)	
Acute lymphoblastic leukemia	97	58 (9.1)	39 (14.2)	
Other hematologic malignancies	236	170 (26.6)	66 (24.1)	
Hematopoietic stem cell transplant	289	199 (24.8)	90 (21.9)	0.259
High-risk MASCC index score ^b^	762	422 (57.4)	340 (92.6)	<0.001
Profound neutropenia (0.1 × 10^9^/L)	725	464 (59.6)	261(65.1)	0.069
Severe mucositis (grade III–IV)	169	109 (13.7)	60 (14.9)	0.574
Corticosteroid therapy (1 month)	630	402 (51.2)	228 (56.6)	0.079
Previous antibiotic therapy (1 month)	663	446 (56.7)	217 (53.3)	0.259
Prior quinolone prophylaxis	195	134 (16.9)	61 (14.9)	0.377
Prior chemotherapy (1 month)	1033	678 (84.9)	355 (86.6)	0.439
Urinary catheter	205	99 (12.8)	106 (26)	<0.001
Nosocomial acquisition	691	475 (59.2)	216 (52.6)	0.026
Source of bloodstream infection				
Endogenous source	452	329 (41)	123 (29.9)	<0.001
Pneumonia	310	154 (19.2)	156 (38)	<0.001
Catheter-related infection	112	78 (9.7)	34 (8.3)	0.408
Neutropenic enterocolitis	71	52 (6.5)	19 (4.6)	0.191
Skin and soft tissue infection	70	49 (6.1)	21 (5.1)	0.480
Urinary tract	51	33 (4.1)	18 (4.4)	0.828
Other abdominal sources ^c^	45	25 (3.1)	20 (4.9)	0.127
Perineal infection	34	29 (3.6)	5 (1.2)	0.017
Mucositis	24	20 (2.5)	4 (1)	0.083
Unknown origin	15	11 (1.4)	4 (1)	0.785
Gangrenous ecthyma	51	32 (4)	19 (4.7)	0.597
MDRPA ^d^	308	169 (21.1)	139 (33.8)	<0.001
XDRPA ^e^	234	122 (15.3)	112 (27.5)	<0.001

^a^ other comorbidities included chronic renal disease and cerebrovascular disease; ^b^ MASCC: Multinational Association for Supportive Care in Cancer; ^c^ other abdominal sources included cholangitis, peritonitis, and intra-abdominal abscesses; ^d^ MDRPA: multidrug-resistant *Pseudomonas aeruginosa*; ^e^ XDRPA: extensively drug-resistant *Pseudomonas aeruginosa.*

**Table 2 microorganisms-12-00705-t002:** Initial empirical antibiotic therapy and clinical outcomes of *P. aeruginosa* bloodstream infection compared by group.

Characteristics	No Septic Shock *n* = 802 (%)	Septic Shock *n* = 411 (%)	*p*-Value
**Empirical antibiotic therapy**			
*Monotherapy*	514 (65.1)	221 (54.7)	<0.001
*Combination therapy*	276 (34.9)	183 (45.3)	<0.001
Both antibiotics with in vitro activity	213 (27)	120 (29.7)	0.324
β-lactam + aminoglycoside	165 (20.9)	93 (23)	0.403
β-lactam + non-aminoglycoside	46 (5.8)	26 (6.4)	0.678
Combination without β-lactam	1 (0.1)	5 (1.2)	0.019
*Inadequate empirical antibiotic therapy*	130 (16.2)	89 (21.7)	0.020
Carbapenems	44 (33.8)	36 (40.4)	
Piperacillin–tazobactam	39 (30)	23 (25.8)	
Fluoroquinolones	12 (9.2)	3 (3.3)	
Ceftriaxone	6 (4.6)	2 (2.2)	
Aminoglycosides	2 (1.5)	3 (3.3)	
Glycopeptides	2 (1.5)	2 (2.2)	
Antipseudomonal cephalosporins	1 (0.76)	2 (2.2)	
Cotrimoxazole	1 (0.76)	1 (1.1)	
Tigecycline	2 (1.5)	1 (1.1)	
Other	21 (16.1)	16 (17.9)	
Granulocyte colony-stimulating factor	424 (53.6)	198 (48.5)	0.096
Intensive care unit admission	84 (10.5)	304 (74)	<0.001
Invasive mechanical ventilation	45 (5.6)	201 (49.1)	<0.001
Persistent bloodstream infection	61 (7.7)	73 (18.3)	<0.001
Early case fatality rate (7 days)	97 (12.1)	239 (58.2)	<0.001
Overall case fatality rate (30 days)	185 (23.1)	304 (74)	<0.001

**Table 3 microorganisms-12-00705-t003:** Risk factors for overall 30-day case fatality rate in patients with septic shock by univariate and multivariate analysis.

Characteristics	Univariate Adjusted OR (95% CI)	Univariate *p*-Value	Adjusted OR (95% CI)	Multivariate *p*-Value
Mechanical ventilation	9.67 (5.35–17.48)	<0.001	9.00 (4.72–17.18)	<0.001
β-lactam + aminoglycoside	0.26 (0.16–0.42)	<0.001	0.58 (0.32–1.05)	0.072
MDRPA ^a^	3.27 (1.87–5.70)	<0.001	1.98 (1.00–3.90)	0.048
Age	1.00 (0.98–1.01)	0.653	1.02 (1.00–1.03)	0.109
Sex	1.04 (0.65–1.66)	0.874	1.25 (0.71–2.22)	0.443
High risk MASCC index score ^b^	0.99 (0.41–2.42)	0.989	0.86 (0.28–2.59)	0.782
IEAT ^c^	4.49 (2.08–10.00)	<0.001	2.47 (1.01–5.88)	0.047
Persistent bloodstream infection	2.54 (1.25–5.15)	0.010	2.49 (1.12–5.56)	0.026
Granulocyte colony-stimulating factor	0.43 (0.27–0.68)	<0.001	0.33 (0.19–0.58)	<0.001
Pneumonia	1.29 (0.81–2.04)	0.286	0.84 (0.39–1.82)	0.652
Urinary tract infection	0.33 (0.13–0.86)	0.023	0.27 (0.07–0.99)	0.047
Endogenous source	0.70 (0.44–1.13)	0.143	0.46 (0.20–1.03)	0.058
Catheter infection	1.16 (0.51–2.64)	0.728	0.65 (0.21–2.05)	0.466

^a^ MDRPA: multidrug-resistant *Pseudomonas aeruginosa*; ^b^ MASCC: Multinational Association for Supportive Care in Cancer; ^c^ IEAT: inadequate empirical antibiotic therapy.

## Data Availability

Data are contained within the article.

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
