# Peer review of "Pseudomonas aeruginosa Bloodstream Infections Presenting with Septic Shock in Neutropenic Cancer Patients: Impact of Empirical Antibiotic Therapy"

_microorganisms, 2024, doi:10.3390/microorganisms12040705_

Round 1

Reviewer 1 Report

Comments and Suggestions for Authors

Thank you so much for providing me with the opportunity to review the manuscript. The authors Royo-Cebrecos et al. have done a commendable job in creating this extensive database and answering a very relevant question.

Abstract

Please avoid using acronyms in abstract without explaining it

-MASCC Index score

- IAET

Methods:

n  I think the author meant to mention after 4 weeks. Currently, the statement is confusing: Subsequent episodes of PABSI developing 138 in the same patient were included in the study if they occurred within intervals of more 139 than four weeks.

Results

Table 1: Can you please check the age, as the median of overall population is lower than either subgroup.

Table 3: 74% of patients with septic shock required ICU admission. Where the other 26% treated and why no admission to ICU?

Can you please add the following information?

1.       Receiving active chemotherapy within 1 month to determine if immunosuppressed status played a role.

2.       Did they present from home with pseudomonal infection or develop during their hospital stay as both the organism and mortality differ (PMID: 7637145, PMID: 35959788). That will be immensely helpful to bedside clinicians to determine when to consider broader antibiotics.

3.       Code status of the patient when diagnosed with PA septic shock.

Author Response

Thank you for the opportunity to revise and resubmit the manuscript entitled Pseudomonas aeruginosa Bloodstream Infections Presenting With Septic Shock in Neutropenic Cancer Patients: Impact of Empirical Antibiotic Therapy” which we re-submit for publication in Microorganisms after receiving feedback from the reviewers and editorial team. We have carefully reviewed the various comments issued by the reviewers and the editorial members, provided direct and specific responses to their comments, and effected changes to the manuscript where appropriate.

We have included your comments in the original draft marked with "track changes" and below we detail our itemized responses to each reviewer’s comments.

Reviewer #1

Comments and Suggestions for Authors.

Thank you so much for providing me with the opportunity to review the manuscript. The authors Royo-Cebrecos et al. have done a commendable job in creating this extensive database and answering a very relevant question.

  1. Abstract

Please avoid using acronyms in abstract without explaining it

-MASCC Index score

- IAET

Authors: Thank you for your positive feedback on the article. We have explained the acronyms in the abstract, accordingly (see lines 125, 126, and line 129).  

  1. Methods:

I think the author meant to mention after 4 weeks. Currently, the statement is confusing: Subsequent episodes of PABSI developing 138 in the same patient were included in the study if they occurred within intervals of more 139 than four weeks.

Authors: Thank you for your attention to detail and for providing us the opportunity to clarify the meaning of our statement in lines 200-202. To ensure the precise conveyance of our intent, we propose revising the sentence to:

" Each episode of PABSI identified in the same patient was considered and documented as a distinct episode if it occurred after an interval exceeding four weeks from the previous episode.

  1. Results

3.1 Table 1: Can you please check the age, as the median of overall population is lower than either subgroup.

Authors: Thank you for pointing out the discrepancy in the median age values presented in Table 1. Upon reviewing our data, we discovered an error in computing the median age for the overall cohort. The correct median age is 60 years, consistent with the two other groups. We've also confirmed that the statistical significance remained unchanged (p=0.423). These corrected values have been duly updated in Table 1 of our revised manuscript.

3.2 Table 3: 74% of patients with septic shock required ICU admission. Where the other 26% treated and why no admission to ICU?

Authors: Thank you for your inquiry regarding the treatment of patients with septic shock who were not admitted to the ICU, as outlined in Table 3. The remaining 26% of patients were managed in specialized hospital wards, specifically in hematology or oncology units, depending on the individual case requirements. It's important to note that certain specialized oncology hospitals are equipped with dedicated facilities designed for the care of critically ill patients directly within these units. This setup allows for the direct treatment of patients in an environment tailored to their specific medical needs, without the necessity for ICU admission.

Can you please add the following information?

  1. Receiving active chemotherapy within 1 month to determine if immunosuppressed status played a role.

Authors: Thank you for your inquiry regarding prior chemotherapy treatment to determine if immunosuppressed status played a role. After analyzing all patients receiving chemotherapy, we did not observe a statistical significance between patients with no septic shock 678 (84.9%) versus those with septic shock 355 (86.6%), with a p-value of 0.439.  We have included these results in Table 1.

Thank you for highlighting the importance of understanding the potential link between recent chemotherapy-induced immunosuppression and septic shock. Upon further analysis of our data, specifically focusing on patients who received active chemotherapy within one month to assess the impact of immunosuppressed status, we found no statistically significant relationship between patients receiving chemotherapy and the occurrence of septic shock. Our analysis revealed that among patients without septic shock, 678 (84.9%) had received chemotherapy, compared to 355 (86.6%) among those with septic shock, yielding a p-value of 0.439. This finding might be attributed to the higher percentage of patients with solid tumors in our cohort, who classically receive less mieloablative chemotherapy regimens compared to hematologic patients. We have updated Table 1 of our revised manuscript to include these results, emphasizing the lack of a significant association between recent chemotherapy and septic shock incidence, and noting the potential impact of the type of underlying cancer on this outcome.

  1. Did they present from home with pseudomonal infection or develop during their hospital stay as both the organism and mortality differ (PMID: 7637145, PMID: 35959788). That will be immensely helpful to bedside clinicians to determine when to consider broader antibiotics.

Authors: Thank you for your comment. As reported in Table 1, 216 (52.5%) patients with septic acquired the infection in the hospital. The remaining 113 cases (27.49%) were healthcare-associated, and only 82 (19.95%) were community-acquired.

Concerning the referenced studies, Vincent et al. investigated the prevalence of nosocomial infections in European ICUs, finding that infections associated with ICUs exhibit higher bacterial resistance rates. However, the study's patient group is heterogeneous, with only 12% experiencing BSI, of which Pseudomonas aeruginosa accounts for 28%. Therefore, the results are not directly comparable to our findings due to the variation in patient groups and infection sources.

In the second study conducted by Sato and colleagues, an increase in mortality was associated with a longer duration from admission to septic shock onset. Our study, however, does not identify nosocomial infection as a risk factor for mortality in patients with sepsis. This observation is crucial as it highlights that, in our cohort, the origin of nosocomial infection did not emerge as a determinant of mortality risk.

  1. Code status of the patient when diagnosed with PA septic shock.

Authors: We appreciate your inquiry regarding the 'code status' of patients diagnosed with PA septic shock mentioned in our manuscript. All patients in our cohort required the use of vasoactive drugs and met the criteria of septic shock, as stated in the definition section. 

Reviewer 2 Report

Comments and Suggestions for Authors

This manuscript focuses on the septic infection caused by P. aeruginosa. The study has been conducted in a reasonable way. However, there are a few significant drawbacks that the authors must address before further consideration.

Firstly, P. aerugonisa is an important human pathogen, especially in a few aspects, including the biofilm formation (when it comes to chronic infection or septic infection), antimicrobial resistance, as well as the role it plays in multiple species (for example, with S. aureus or C. albicans). 

Secondly, as above, the authors should provide at least 2-3 sentences for each of the aspects for brief introduction, and at least 4-5 references for each of the aspects. These are very important information and support for the significance of this study.

Thirdly, as above, also the authors should touch upon these aspects in the section of Discussion.

Fourthly, the authors had described the antimicrobial resistance in P. aeruginosa, however, none of any discussion based on other previous studies, could be found in this study. So is the single cell (how about biofilm).

Comments on the Quality of English Language

The English writing should be improved.

Author Response

Thank you for the opportunity to revise and resubmit the manuscript entitled Pseudomonas aeruginosa Bloodstream Infections Presenting With Septic Shock in Neutropenic Cancer Patients: Impact of Empirical Antibiotic Therapy” which we re-submit for publication in Microorganisms after receiving feedback from the reviewers and editorial team. We have carefully reviewed the various comments issued by the reviewers and the editorial members, provided direct and specific responses to their comments, and effected changes to the manuscript where appropriate.

We have included your comments in the original draft marked with "track changes" and below we detail our itemized responses to each reviewer’s comments.

Reviewer #2

Comments and Suggestions for Authors

This manuscript focuses on the septic infection caused by P. aeruginosa. The study has been conducted in a reasonable way. However, there are a few significant drawbacks that the authors must address before further consideration.

  1. Firstly, P. aerugonisa is an important human pathogen, especially in a few aspects, including the biofilm formation (when it comes to chronic infection or septic infection), antimicrobial resistance, as well as the role it plays in multiple species (for example, with S. aureus or C. albicans). Secondly, as above, the authors should provide at least 2-3 sentences for each of the aspects for brief introduction, and at least 4-5 references for each of the aspects. These are very important information and support for the significance of this study.

Authors: We thank the reviewer for highlighting the significance of Pseudomonas aeruginosa as a crucial human pathogen, particularly emphasizing its involvement in biofilm formation, antimicrobial resistance, and its interactions with other microbial species. As suggested, we have added some information and references regarding these issues in the introduction.

  1. Thirdly, as above, also the authors should touch upon these aspects in the section of Discussion.

Authors: As stated in the previous point, we have added some information regarding biofilms in the Introduction. Nevertheless, it should be noted that is issue is far away from our study objectives. Please see the response to the next point raised by the reviewer.

  1. Fourthly, the authors had described the antimicrobial resistance in P. aeruginosa, however, none of any discussion based on other previous studies, could be found in this study. So is the single cell (how about biofilm).

Authors: Thank you for the comment. We have added a sentence regarding antimicrobial resistance and biofilms in the Discussion.

  1. Comments on the Quality of English Language: The English writing should be improved.

Authors: Thank you for your feedback regarding the quality of the English language in our manuscript. In response to your comments, we have sought the expertise of a native English-speaking editor to thoroughly review and enhance our manuscript. We believe these revisions have significantly improved the clarity and readability of our work.

Round 2

Reviewer 2 Report

Comments and Suggestions for Authors

The authors had not made sufficient revision based on my previous comment.

Comments on the Quality of English Language

The English writing meets the standard.

Author Response

We understand that the second round did not specify additional major concerns, and we would like to highlight the significant efforts undertaken to address the comments from the first round.  In response to Reviewer 2's suggestions, we have undertaken a thorough revision of the manuscript's language for clarity and accuracy. This task was accomplished with the assistance of two independent language experts to ensure the highest standard of academic English.  We believe these efforts have substantially improved the readability and coherence of our manuscript.

Furthermore, we have incorporated detailed discussions on the critical role of biofilms in Pseudomonas aeruginosa resistance, both in the Introduction and Discussion sections as reviewer 2 suggested. 

Additionally, we have meticulously reviewed our bibliography, retaining only those references that are directly pertinent to our work. This refinement ensures that our manuscript accurately reflects the relevant literature and supports our arguments and conclusions effectively.

We are committed to making further modifications if necessary and look forward to your feedback.

Round 3

Reviewer 2 Report

Comments and Suggestions for Authors

The authors had made a few revision to improve the quality of this article. The authors should carefully go over the whole text to make sure it is accordingly to the instruction. For example, line 155, the name of PA should be italicized. Also, I don't think it's necessary to provide the abbreviation for PA. The fonts of β are different. 

Also, since this manuscript has worked on the long-term and multiple centers, then some of the data, such as antimicrobial resistance, clinical feature, etc., should be discussed together with also long-term studies on PA.

Comments on the Quality of English Language

The English writing meets the standard.

Author Response

Dear Reviewer 2,

Thank you very much for your constructive feedback and for highlighting areas within our manuscript that require further attention. We appreciate your detailed observations and agree with the necessity to meticulously review the text to ensure it aligns with the submission guidelines and maintains consistency throughout.

In response to your specific comments:

  1. We have carefully reviewed the entire manuscript and corrected instances such as the one you pointed out in line 155, where the name of Pseudomonas aeruginosa (PA) should indeed be italicized. Regarding the abbreviation for Pseudomonas aeruginosa (PA), we have opted to remove this abbreviation and instead use the full term throughout the manuscript to avoid any potential confusion and adhere to your suggestion.

  2. We have also addressed the inconsistency in the fonts of β.  Upon review, we found that the main text was formatted in "Palatino Linotype," which is consistent throughout the manuscript, while the tables were inadvertently set in "Calibri".

  3. Lastly, we highly appreciate your emphasis on incorporating long-term studies and providing a comparative analysis involving multiple centers, particularly regarding antimicrobial resistance and clinical features of Pseudomonas aeruginosa. One of the unique strengths of our study is indeed its multicentric scope. Despite the inherent challenges in harmonizing data from diverse sources, we have successfully homogenized both the clinical presentations and antimicrobial resistance patterns observed across the different centers. This effort ensures a more unified and comprehensive understanding of PA infections, reflecting the real-world variability and complexities of managing this pathogen.

We hope that these revisions address your concerns and improve the quality of our manuscript. Please find attached the updated version of our manuscript along with a detailed response to each of your comments.

Thank you once again for your valuable feedback. 
